# Online PCA in Converging Self-consistent Field Equations

**Xihan Li**[1], **Xiang Chen**[2], **Rasul Tutunov**[3], **Haitham Bou Ammar**[3], **Lei Wang**[4], **Jun Wang**[1]

[1] University College London    [2] Huawei Noah's Ark Lab    [3] Huawei R&D U.K.

[4] Institute of Physics, Chinese Academy of Sciences

`{xihan.li,jun.wang}@cs.ucl.ac.uk`

`{xiangchen.ai,rasul.tutunov,haitham.ammar}@huawei.com`

`wanglei@iphy.ac.cn`

## Abstract

Self-consistent Field (SCF) equation is a type of nonlinear eigenvalue problem in which the matrix to be eigen-decomposed is a function of its own eigenvectors. It is of great significance in computational science for its connection to the Schrödinger equation. Traditional fixed-point iteration methods for solving such equations suffer from non-convergence issues. In this work, we present a novel perspective on such SCF equations as a principal component analysis (PCA) for non-stationary time series, in which a distribution and its own top principal components are mutually updated over time, and the equilibrium state of the model corresponds to the solution of the SCF equations. By the new perspective, online PCA techniques are able to engage in so as to enhance the convergence of the model towards the equilibrium state, acting as a new set of tools for converging the SCF equations. With several numerical adaptations, we then develop a new algorithm for converging the SCF equation, and demonstrated its high convergence capacity with experiments on both synthesized and real electronic structure scenarios.

## 1 Introduction

In this work, we are concerned with the convergence issue of solving the following type of nonlinear eigenvalue problem:

$$F(v)v = \lambda v \tag{1}$$

in which $F(v)$ is a given mapping from an $N$-dimensional unit vector to an $N \times N$ positive semi-definite matrix, $\lambda$ is the largest eigenvalue of $F(v)$, and $v$ is the normalized eigenvector of $F(v)$ corresponding to $\lambda$. The core challenge of this problem is the involvement of *self-consistency*, that is, the form of the eigenvalue equation (i.e., the matrix $F(v)$ to be decomposed) is defined by its final solution $v$, but the solution itself is not directly accessible unless the form of the equation is given, which is a paradox. Such type of nonlinear eigenvalue problem is especially concerned in computational science, since some variations of Equation (1) like Hartree-Fock equations [10, 11] and Kohn-Sham equations [15, 19] are the foundation of electronic structure calculation by approximating the Schrödinger equation, in which $F(v)$ corresponds to the Fock operator approximating the Hamiltonian operator of a quantum system, and the eigenvector $v$ corresponds to the coefficients of an orbital wave function under certain basis [33, 25]. Such equations are usually called "self-consistent field (SCF) equations".

However, like many other types of nonlinear equations, no existing numerical algorithms can solve SCF equations with theoretical convergence guarantee, while many works are committed to improve the practical "successful rate" for the solution to be converged. To be more specific, the iterative

method to solve SCF equations is generally referred to as self-consistent field (SCF) method [30, 23], whose basic idea is to regard the SCF equation as a fixed-point equation $v = f(v)$ so as to perform fixed-point iteration. For Equation (1), we can rewrite it as $v = \text{Eig}(F(v))$ in which $\text{Eig}(\cdot)$ returns the eigenvector of a matrix corresponding to its largest eigenvalue. Then the fixed-point iteration is performed by generating an initial solution $v_0$ and performing $v_{t+1} = \text{Eig}(F(v_t))$ iteratively until convergence. Note that there is no theoretical guarantee that the above iteration step leads to a converged solution[1]. In practice, vanilla SCF method also easily fails in real electronic structure calculation under Hartree-Fock or Kohn-Sham equations, acting as oscillating between two or more different states that are not solutions of the equations [4]. To mitigate the convergence issue of SCF method, existing works mainly follow two directions: one is to generate better-quality initial solutions $v_0$ in a semi-empirical way [14, 35, 22], another one is to mix $F(v_t)$ with those in previous iterations $F(v_{t-1}), F(v_{t-2}), \cdots$ to stabilize the iteration process [28, 21, 16, 6].

In this work, we propose a different direction to converge SCF equations. We have an insight that the essence of SCF equation is a special type of eigen-decomposition where the matrix to be decomposed is not determined during the decomposition. In this way, it shares a key similarity with principal component analysis (PCA) in a non-stationary time-series environment, as PCA is also rooted on eigen-decomposition, and the covariance matrix of the data distribution is not determined as it may change over time. By transiting to the PCA perspective, it is possible to apply a series of online PCA techniques that are successfully developed to handle PCA in non-stationary time-series (or streaming) environments. They are able to adapt to new patterns in the data stream dynamically by incremental updates of the principal component. Connecting it with SCF equations, such an incremental feature is potentially helpful in mitigating the convergence issue, especially by preventing long-range oscillation between successive iterations. Motivated by these insights, we propose a dynamic PCA model with an auto-encoder structure, whose equilibrium state is the solution of Equation (1). In this model, we view the eigenvector $v$ in the SCF equation as a principal component for a certain data distribution $\mathcal{P}(x; \Sigma)$ in PCA, and interpret $F(v)$ in the SCF equation as a "reconstruction function" that convert a vector $v$ to the covariance matrix $\Sigma$ of a certain distribution. Then we utilize online learning techniques to lead the dynamic PCA model towards equilibrium state.

For some important real-world applications of SCF equations, such as Hartree-Fock and Kohn-Shan equations for electronic struction calculation, we also show that our PCA-based model can be adapted to these applications, and proposed several numerical adaptations to converge these more complicated equations. Particularly, we proposed an adaptive mechanism that allow the iteration to switch between online mode for convergency and regular mode for accuracy, which provides unlimited chances of trials to reach a converged trajectory, which is lacking in standard SCF methods when stuck in oscillation.

Experimental result on synthesis and real scenarios shows that our proposed approach largely reduce the occurrence of oscillation, leading to a significant improvement of convergence performance. In this way, our work expands the reach of online PCA methods into handling self-consistency. In summary, we make the following contributions:

- A novel formulation of the SCF equation as finding the equilibrium state of a dynamic PCA model for non-stationary time series.
- A new application of online learning techniques to the proposed model so as to enhance its convergence to the equilibrium state and avoid oscillation, improving the convergence performance of SCF equations solving in a generic way.
- A new algorithm based on online PCA with several numerical adaptations, which is capable of converging the SCF equation in real-world electronic structure calculation with high successful rate.
- Extensive experiments on a synthetic problem and real datasets for electronic structure calculation, demonstrating the high capacity of the proposed algorithm in converging the SCF equation.

---

[1]To have an insight, consider that the mapping $\text{Eig}(F(\cdot)) : \mathbb{R}^N \to \mathbb{R}^N$ is generally not a contraction mapping so that the Banach fixed-point theorem [2] does not work here.

## 2 Related work

With the prosperity of deep learning and differentiable optimization in recent years, there are works combining machine learning with computational science and electronic structure calculation, including deep-learning-aided wave function representation of quantum Monte Carlo methods such as FermiNet [27], PauliNet [13] and [3] and neural representation of the exchange-correlation functional in density functional theory [24, 18]. However, these works stay within traditional formalism of the problem, focusing more on improving simulation accuracy towards physical reality by using neural networks as better functional approximators, while our work's focus is very different, stressing on a machine learning oriented formalism of the SCF equation, which has not been explored in the literature to the best of our knowledge.

To converge SCF equations, existing works mainly follow two directions. One is to generate better-quality initial solutions $v_0$ [14, 35, 22]. However, these methods are generally semi-empirical as they require specific assumption of $F$, and leverage domain knowledge (e.g., quantum mechanism) for the initialization. another one is to mix $F(v_t)$ with those in previous iterations $F(v_{t-1}), F(v_{t-2}), \cdots$ to stabilize the iteration process [28, 21, 16, 6]. While these methods perform well on converging SCF equations efficiently, they can still be stuck in indefinite oscillating between non-solution states without the chance of escaping from it.

Principal component analysis (PCA) is a fundamental, well-studied tool used to for data analysis and compression [17]. The principal components, which are "representative directions" of the data distribution that preserve the data's variation, can be computed by eigen-decomposition of the data covariance matrix. However, when the data are formed as an online, non-stationary stream whose distribution (or more specifically, the covariance matrix) may shift over time, specialized online learning techniques are required to estimate the top-$k$ principal components in a real-time manner, and dynamically adapt to new patterns in the data stream. Starting from Hebb's rule in neuroscience [12], a series of works [26, 34, 1, 7] focus on this direction named Online $k$-PCA.

## 3 Solving SCF Equations with Online PCA

To mitigate the convergence issue in the solving of the SCF equation (1), we connect it with PCA by proposing a new PCA model for non-stationary time series, showing that the solving of the SCF equation is equivalent to finding the equilibrium state of the proposed PCA model. In this way, online learning techniques for PCA can be exploited to accelerate the convergence. We also propose some numerical adaptations so as to solve real-world SCF equations in electronic structure computation.

### 3.1 The Connection between SCF Equations and PCA

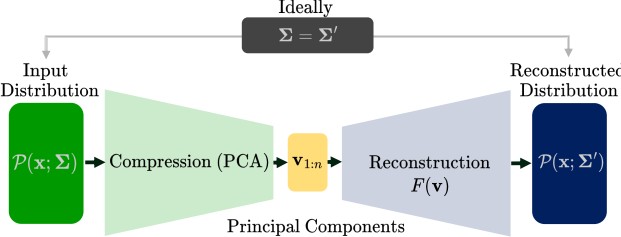

Figure 1: SCF equations as finding a distribution that is invariant before and after being processed by an auto-encoder structure involving PCA. An input distribution $\mathcal{P}(\boldsymbol{x}; \boldsymbol{\Sigma})$ is compressed yielding a set of principal components $\boldsymbol{v}_{1:n}$. Those are then used to produce a reconstructed distribution $\mathcal{P}(\boldsymbol{x}; \boldsymbol{\Sigma}')$, where ideally we would like $\boldsymbol{\Sigma} = \boldsymbol{\Sigma}'$.

To connect the SCF equation with PCA, notice that Equation (1) can also be stated as

$$v = \text{Eig}(F(v)) \tag{2}$$

or equivalently

$$\Sigma = F(\text{Eig}(\Sigma)) \tag{3}$$

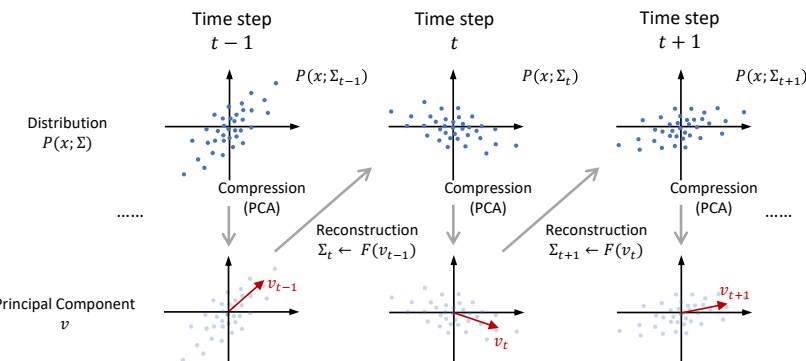

Figure 2: A diagram of the dynamic PCA model, in which a distribution $\mathcal{P}(\boldsymbol{x}; \boldsymbol{\Sigma})$ and its top principal component $\boldsymbol{v}_{1:n}$ are mutually updated over time. Importantly, we notice that we can regard the approach as applying PCA on non-stationary distributions that evolve over time. Given that such a formalism is novel and is an outcome of SCFs, the machine learning literature is yet to devise effective solutions. In this work, we take the first steps in devising such algorithms and demonstrate successful applications in SCFs.

where $\text{Eig}(\cdot)$ is the eigenvector of a matrix corresponding to its largest eigenvalue, and $v = \text{Eig}(\Sigma)$. Here, if we consider the function $\Sigma = F(v)$ as a "reconstruction function" that convert a vector $v$ to the covariance matrix $\Sigma$ of a certain data distribution $\mathcal{P}(x; \Sigma)$, then the function $v = \text{Eig}(\Sigma)$ is equivalent to a "compression function" that finds the top principal component $v$ of this distribution $\mathcal{P}(x; \Sigma)$ with PCA, since the top principal component is exactly the eigenvector of the distribution's covariance matrix corresponding to the largest eigenvalue. From this perspective, the solving of Equation (3) (and equivalently (2)) can be seen as finding a distribution $\mathcal{P}(x; \Sigma)$ parameterized by the covariance matrix $\Sigma$ that is invariant before and after being processed by the following two stages:

- Compression (PCA): perform PCA on the distribution $\mathcal{P}(x; \Sigma)$ so as to obtain its top principal component $v$, which is the representative direction of the distribution.
- Reconstruction ($F(v)$): process $v$ with the given reconstruction function $\Sigma' = F(v)$ so as to obtain the reconstructed distribution $\mathcal{P}(x; \Sigma')$.

which is shown in Figure 1. While its architecture seems similar to the autoencoder [20], the encoder (PCA) and decoder (reconstruction function $F(v)$) here are both fixed without any adjustable parameters. Instead of training the encoder and decoder, here we aim to find an input distribution $\mathcal{P}(x; \Sigma)$ which remains invariant after been "encoded" and "decoded".

## 3.2 A New PCA Model for Non-stationary Time Series

For solving of the SCF equation (2), traditionally we perform fixed-point iteration by letting the fixed-point mapping $f(v) = \text{Eig}(F(v))$ and performing $v_{t+1} = f(v_t)$ iteratively until convergence. Here, by replacing $f$ with the reconstruction and compression stages mentioned above, we obtain an equivalent form of the fixed-point iteration, which is presented as a dynamic model that a distribution $\mathcal{P}(x; \Sigma)$ and its top principal component $v$ are mutually updated over time, shown in Figure 2. The evolution process of the dynamic PCA model is as follows:

> Given initial top principal component $v_0$, reconstruction function $F(v)$
> For $t = 1, 2, \cdots$ until converge ($\|v_t - v_{t-1}\| < \epsilon$)
> - Reconstruct the distribution $\mathcal{P}(x; \Sigma_t)$ by $\Sigma_t = F(v_{t-1})$.
> - Perform PCA on $X_t \sim \mathcal{P}(x; \Sigma_t)$ and obtain the top principal component $v_t \leftarrow \text{PCA}(X_t)$.

In this model, the top principal component is updated by performing PCA on the current distribution, and the distribution is updated by performing reconstruction on the current top principal component. As it is equivalent to the fixed-point iteration, the equilibrium state of the new dynamic model is also the fixed point of Equation (2), which corresponds to the solution of Equation (1).

An important feature of the dynamic PCA model is that, it can be regarded as applying PCA on a non-stationary distribution over time, as the distribution that is processed by PCA is subject to change

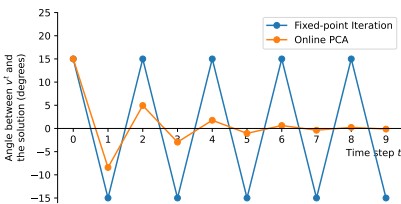

Figure 3: Convergence comparison between vanilla fixed-point iteration and Online PCA given $F(v) = \begin{bmatrix} v_2^2 & v_1 v_2 \\ v_1 v_2 & v_1^2 \end{bmatrix}$, if we set the initial solution $v_0$ as $[1/2, \sqrt{3}/2]^\top$. The principal component at time step 0 is set to be $[1/2, \sqrt{3}/2]^\top$ for both models. While the vanilla fixed-point iteration oscillates between $[1/2, \sqrt{3}/2]^\top$ and $[\sqrt{3}/2, 1/2]^\top$ along with the time step, the dynamic PCA model proposed in Section 3.2 converges to the ground-truth solution $[\sqrt{2}/2, \sqrt{2}/2]^\top$ steadily after applying online PCA technique.

over time in the evolving process. This new perspective enables a variety of PCA techniques for non-stationary environments to be applied on the solving of the problem, whose details are discussed below.

### 3.3 Online PCA for Converging SCF Equations

As we have discussed in the introduction, vanilla fixed-point iteration method has serious convergence issues. acting as oscillating between two or more different points, neither of which are the solution of Equation (2). For example, given $F(v) = \begin{bmatrix} v_2^2 & v_1 v_2 \\ v_1 v_2 & v_1^2 \end{bmatrix}$, if we set the initial solution $v_0$ as $[1/2, \sqrt{3}/2]^\top$ and perform fixed-point iteration $v_k = \mathrm{Eig}(F(v_{k-1}))$, we will find it oscillating between $[1/2, \sqrt{3}/2]^\top$ and $[\sqrt{3}/2, 1/2]^\top$ along with the time step. Neither of them are close to the analytical solution $[\sqrt{2}/2, \sqrt{2}/2]^\top$. However, the new PCA-based perspective allows us to apply online learning techniques to mitigate this issue. In online PCA, the principal components are usually updated in an incremental manner so as to adapt to new patterns in the data distribution. The incremental property is especially appealing to us, since the oscillation of fixed-point iteration behaves as infinite "jumps" between different states, and incremental updates can reduce such jumps by adding soft restrictions to the difference of principal components between successive time steps. After applying online learning techniques, the evolution process of the dynamic PCA model is as follows:

Given initial top principal component $v_0$, reconstruction function $F(v)$
For $t = 1, 2, \cdots$ until converge ($\|v_t - v_{t-1}\| < \epsilon$)
    • Reconstruct the distribution $\mathcal{P}(x; \Sigma_t)$ by $\Sigma_t = F(v_{t-1})$.
    • Sample $x_t$ from $\mathcal{P}(x; \Sigma_t)$
    • Update $v$ by online PCA with $x_t$

and an illustrative example is shown in Figure 3.

### 3.4 A Case Study for the Convergence Behavior of Online PCA

For the iterative methods solving Equation (1), while the convergence analysis is generally intractable due to the arbitrariness of $F(v)$ (usually nonlinear by involving $vv^\top$), here we provide a case study for a specific form of $F(v)$ whose analytical ground truth solutions are available. While vanilla fixed-point iteration method doesn't work in this case, we can derive and visualize the convergence behavior of our proposed online PCA method in an analytical way. Here we let

$$F(v) = (Av)(Av)^\top = Avv^\top A^\top \tag{4}$$

in which $A$ is an orthogonal matrix. Substitute Equation (4) into (1) and we obtain

$$(Avv^\top A^\top)v = \lambda v \tag{5}$$

The analytical solution of Equation (5) is the normalized eigenvector of $A$ corresponding to the eigenvalue 1, since the largest eigenvalue of matrix $Avv^\top A^\top$ is 1 with corresponding eigenvector

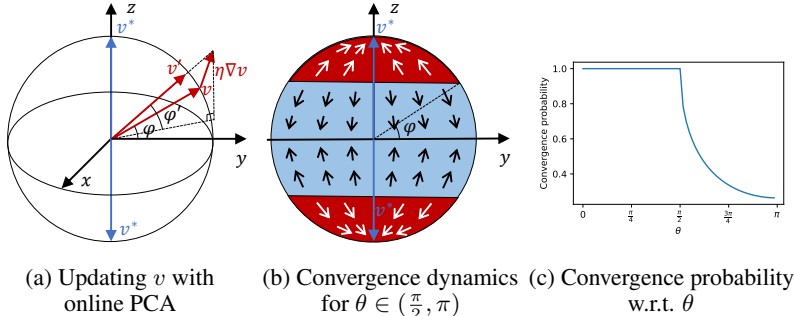

(a) Updating $v$ with online PCA

(b) Convergence dynamics for $\theta \in (\frac{\pi}{2}, \pi)$

(c) Convergence probability w.r.t. $\theta$

Figure 4: Convergence analysis of our proposed online PCA method for solving Equation (5). Note that vanilla fixed-point iteration can never converge in this case.

$Av$ (i.e., $\text{Eig}(Avv^\top A^\top) = Av$), reducing Equation (5) to a standard eigenvalue problem $v = Av$. However, The vanilla fixed-point iteration method cannot converge when solving Equation (5). Note that the mapping in each iteration will be reduced to $v_{t+1} = Av_t$, which seems very similar to the power method on $A$ for finding the eigenvector with the largest eigenvalue, but with convergence ratio $|\frac{\lambda_2}{\lambda_1}| = 1$ (i.e., cannot converge) since all the absolute value of $A$'s eigenvalues are 1 as $A$ is an orthogonal matrix.

Now we turn to the convergence analysis of our proposed online PCA method for this case. For the sake of visualization, we further restrict $A$ to be three-dimensional as a rotation transformation matrix around the $z$-axis with angle $\theta$, that is

$$A = \begin{bmatrix} \cos\theta & -\sin\theta & 0 \\ \sin\theta & \cos\theta & 0 \\ 0 & 0 & 1 \end{bmatrix} \tag{6}$$

so that the solution of Equation (5) will be $v^* = [0, 0, \pm 1]^\top$, the two unit vectors on the $z$-axis. Then, we let the initial top principal component $v_0$ to be a unit vector on the $yz$ plane

$$v_0 = [0, \cos\varphi_0, \sin\varphi_0]^\top, \varphi_0 \in [0, \frac{\pi}{2}] \tag{7}$$

where $\varphi_0$ is the initial angle between the vector and the $xy$ plane.

We analytically derived the convergence behavior of the online PCA method described in Section 3.3 for solving Equation (5), whose detail is leaved in the appendix. The result is as follows: for $\theta \in (0, \frac{\pi}{2})$, the online PCA method is guaranteed to converge to the ground-truth solution on $z$-axis. However, for $\theta \in [\frac{\pi}{2}, \pi)$, there is a phase transition of convergence behavior w.r.t. the initial angle $\varphi_0$, shown in Figure 4b. While the online PCA method stays converged for $\varphi_0 > \arccos\sqrt{x_1(\theta)}$, it will fail to converge otherwise. Assuming the initial vector is sampled uniformly on the unit sphere, the probability of convergence for the online PCA method w.r.t. the rotation angle $\theta$ is shown in Figure 4c.

### 3.5 Numerical Adaptations for Converging Real-world SCF Equations

Now we investigate a more complicated and applicative scenario in electronic structure calculation, which is usually known as Hartree-Fock equations or Kohn-Sham equations. The form of such SCF equations [33] is

$$F(V)V = SV\Lambda, \tag{8}$$

where

- $F(V)$: an $N \times N$ real symmetric matrix to be decomposed, as a given function of $V$. It is defined as follows:

$$F(V) \stackrel{\text{def}}{=} H + U_{\text{eff}}(2VV^\top), \tag{9}$$

  in which $H$ is an $N \times N$ real symmetric matrix and $U_{\text{eff}}(\cdot)$ is an $N \times N$ real symmetric matrix as a function of $2VV^\top$. Both of them are given in the equation.
- $S$: an $N \times N$ positive semi-definite matrix, which is a constant input in the problem.

- $\Lambda$: $\Lambda = \mathrm{diag}(\lambda_1, \cdots, \lambda_k)$ is a $k \times k$ diagonal matrix containing the top-$k$ smallest eigenvalues.
- $V$: $V = [v_1, \cdots, v_k]$ is an $N \times k$ matrix containing $k$ normalized column eigenvectors corresponding to the top-$k$ smallest eigenvalues.

Our proposed model can be adapted to handle it as follows

1. While $F(v)$ is not guaranteed to be positive semi-definite, note that we can always make an "eigenvalue shift" towards $F(v)$ by adding it with $\sigma I$ ($\sigma$ is a moderately large number) to make it positive semi-definite, without changing its eigenvectors (i.e., the solution of Equation (8)).
2. For $S \neq I$, Equation (1) becomes $F(v)v = \lambda Sv$ which is a generalized eigen-decomposition problem[8]. We can then perform standard orthogonalization technique that transforms the generalized eigenvalue problem into a standard one [33].
3. To tackle the smallest eigenvalues instead of the largest ones, we can simply add a minus sign to $F(v)$, so that the order of its eigenvalues will be reversed without changing its eigenvectors.
4. For $k > 1$, our proposed model can be trivially extended to $k > 1$ cases by retaining $k$ top principal components rather than one.

And we name the adapted model as "Online SCF". Then, we introduce some practical improvements that can further enhance the efficiency and convergence capability of our proposed model for solving Equation (8):

**Adaptive Update Interval**: Instead of updating $\Sigma$ in every iteration, we control the update interval of $\Sigma$ via a parameter $I_\Sigma$ to improve efficiency. Since $V$ is only updated slightly in each iteration step with a small learning rate $\eta$, it may not be necessary to do a fresh computation of $\Sigma = F(V)$ in each time step, especially considering that the computation of $U_{\mathrm{eff}}(\cdot)$ in Equation (9) can be costly. Moreover, we change $I_\Sigma$ adaptively in the iteration to improve the efficiency by the following intuition: if there is a significant change between $\Sigma_t$ and $\Sigma_{t-1}$, our dynamic PCA model is in an unstable state and we should apply online PCA method in a more responsive way to improve the convergence by setting a smaller $I_\Sigma$. However, if the gap is small, then our dynamic PCA model is very likely to be already close to a stable state, in which case the precision of PCA result is more dominant for obtaining a highly precise solution (especially for electronic structure calculation requiring error $< 10^{-10}$), and a larger $I_\Sigma$ is more reasonable. In this way, we set $I_\Sigma$ to be inversely proportional to the difference between $\Sigma_t$ and $\Sigma_{t-1}$, as $I_\Sigma^t = \lceil \frac{1}{\Delta(\Sigma_t, \Sigma_{t-1})} \rceil$ in which $\Delta(\Sigma_t, \Sigma_{t-1}) \geq 0$ is a properly scaled function evaluating the difference between successive $\Sigma$.

**Adaptive PCA Mode Switching**: Note that, the larger $I_\Sigma$ is, the more costly in time between two updates of $\Sigma$, and the closer the result is to regular PCA on $\mathcal{P}(x; \Sigma)$. Thus a cut-off value $T_{\mathrm{cut\text{-}off}}$ is set so that the PCA model will switch from online to regular when $I_\Sigma^t > T_{\mathrm{cut\text{-}off}}$, avoiding exhausted iteration on a single $\Sigma$. Moreover, this brings the preciseness needed for the final stage of the iteration when the error is small and convergence may not be a problem. Empirically, such a mode switching will introduce temporary disturbance for a few iterations, so we also set a small "tabu tenure"[2] $T_{\mathrm{tabu}}$ prohibiting switching back to the original PCA mode in $T_{\mathrm{tabu}}$ steps. Such switching between online and regular PCA can be triggered multiple times. A high amount of switches indicates that the convergence of the iteration may be hard, so $T_{\mathrm{cut\text{-}off}}$ will increase by a small value $T_{\mathrm{cut\text{-}off\text{-}inc}}$ after each switching to increase the proportion of online PCA to handle the convergence issue.

Additionally, we also applied the direct inversion of the iterative subspace (DIIS) method, which is empirically effective in traditional methods for electronic structure calculation. The basic idea is to update $\Sigma_t$ at iteration $t$ as a linear combination of matrices in previous $T$ iterations $\Sigma_{t-T}, \cdots, \Sigma_{t-1}$, whose detail can be found in [28] and the appendix. In the experiments involving electronic structure calculation, we apply DIIS on all tested methods.

Some other numerical adaptations, including the application of momentum and sample-free update, are also leaved in the appendix. Summarizing all considerations above, we propose an algorithm for solving Equation (8), named "Adaptive Online SCF", shown in Algorithm 1

---

[2]This term is borrowed from Tabu Search [9].

**Algorithm 1** Adaptive Online SCF for solving Equation (8) in electronic structure calculation

---

**Input:** $H, S, U_{\text{eff}}(\cdot)$ in Equation (8) and (9), learning rate $\eta$, difference evaluation function $\Delta_1(\Sigma_t, \Sigma_{t-1})$ for convergence criteria and $\Delta_2(\Sigma_t, \Sigma_{t-1})$ for the computation of $I_\Sigma$, cut-off threshold $T_{\text{cut-off}}$ and increment value $T_{\text{cut-off-inc}}$, tabu tenure $T_{\text{tabu}}$, convergence threshold $\epsilon$

**Output:** $V^*$, the solution of Equation (8)

1: Initialize $V_0' \in \mathbb{R}^{N \times k}$ randomly.
2: $\Sigma_0 \leftarrow H$
3: Find $X$ satisfying $X^\top S X = I$.
4: $s_0 \leftarrow$ Online
5: $t \leftarrow 0, i \leftarrow 0$
6: **while** $\Delta_1(\Sigma_t, \Sigma_{t-1}) > \epsilon$ **do**
7:      $V_t \leftarrow X V_t'$
8:      $\Sigma_t \leftarrow \text{DIIS}(\Sigma_{t-S}, \cdots, \Sigma_{t-1}, F(V))$
9:      $\Sigma_t' \leftarrow X^\top \Sigma_t X$
10:      $I_\Sigma^t = \lceil 1/\Delta_2(\Sigma_t, \Sigma_{t-1}) \rceil$
11:      **if** $I_\Sigma^t \leq T_{\text{cut-off}}$ and not $(s_t = \text{Regular and } i < T_{\text{tabu}})$ or $(s_t = \text{Online and } i < T_{\text{tabu}})$ **then**
12:          **for** $t' = 0, 1, 2, \cdots, I_\Sigma^t$ **do**
13:              Use online PCA to update $V_t'$ towards "covariance matrix" $-\Sigma_t'$, with momentum.
14:          **end for**
15:          $s_t \leftarrow$ Online
16:      **else**
17:          Use regular PCA to compute "principal component" $V_t'$ from "covariance matrix" $-\Sigma_t'$.
18:          $s_t \leftarrow$ Regular
19:      **end if**
20:      $i \leftarrow i + 1$ **if** $s_t = s_{t-1}$ **else** $0$
21:      $T_{\text{cut-off}} \leftarrow T_{\text{cut-off}}$ **if** $s_t = s_{t-1}$ **else** $T_{\text{cut-off}} + T_{\text{cut-off-inc}}$
22:      $t \leftarrow t + 1$
23: **end while**
24: $V^* \leftarrow X V'$

---

## 4 Experiments

### 4.1 Experimental Verification of the Case Study

To verify the convergence analysis in Section 3.4, we conduct Monte Carlo experiments to solve Equation (5) for rotation matrix $A$ with different rotation angle $\theta$. For each angle, we uniformly sample 1,000 initial vectors $v_0$ on the unit sphere, perform the online PCA method and traditional fixed-point iteration (vanilla SCF) starting from these vectors to solve Equation (5), and use the proportion of converged vectors as an approximation of convergence probability. To be aligned with the next section, we also included the DIIS method described in the appendix that is extensively used in solving Equation (8).

The result is shown in Figure 5. While vanilla SCF (fixed-point iteration) cannot converge on any instance, the convergence probability of online PCA is aligned with the analytical result shown in Figure 4c. DIIS achieves around 40% convergence probability regardless of the rotation angle, which is significantly better than vanilla SCF but still fall behind for $\theta < 3\pi/4$ compared with online PCA method. The result validates the convergence analysis in Section 3.4, showing the potentially high convergence capacity of online PCA method.

### 4.2 Convergence Capability Evaluation on Electron Structures

In this section, we perform extensive benchmarks on the QM9 dataset [31, 29], a diverse, large dataset to evaluate the capacity of converging Equation (8), the SCF equation in the scenario of electronic structure calculation. The QM9 dataset contains the atomic coordinates of 133,885 molecules in total, which is huge in size, thus we sampled 1% of the dataset (1,338 molecules) in a purely random manner for our evaluation. The effective potential matrix function $U_{\text{eff}}(\cdot)$ in Equation (9) is based on Hartree-Fock theory and Density Functional Theory (DFT) with B3LYP exchange-correlation functional, provided by PySCF [32]. We evaluated the methods on the two theories separately.

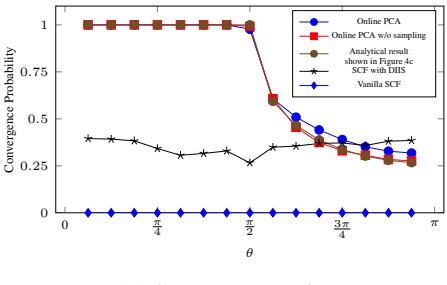
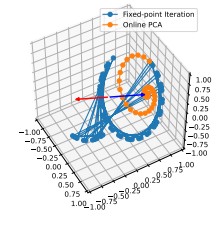

(a) Convergence ratio

(b) An example of
convergence trajectories

Figure 5: Performance evaluation of solving Equation (5) whose analytical solutions are available. (a) While vanilla SCF (fixed-point iteration) cannot converge on any instance, the convergence probability of online PCA is aligned with the analytical result shown in Figure 4c. (b) An example of convergence trajectories, traditional SCF (fixed-point iteration) fails to converge (cycling around the solutions) while our Online PCA method converges to the solutions (red and blue arrow) smoothly.

| Methods | Hartree-Fock | | DFT with B3LYP | |
|---|---|---|---|---|
| | #(Nonconverged molecules) | Average #(iterations) | #(Nonconverged molecules) | Average #(iterations) |
| Regular SCF | 124 (9.27%) | 25.49 | 407 (30.42%) | 21.09 |
| Full Online SCF | 13 (0.97%) | 584.68 | 217 (16.22%) | 1835.24 |
| Adaptive Online SCF | 0 (0%) | 42.97 | 0 (0%) | 60.58 |

Table 1: Results on 1,338 randomly sampled molecules in QM9 dataset. All methods are initialized with core Hamiltonian and accelerated by DIIS. Average #(iterations) is for converged molecules only.

Standard 6-31G basis set [5] are applied for the computation of all molecules. All tested methods are initialized with core Hamiltonian and accelerated by DIIS. Three methods are evaluated as follows:

- **Regular SCF**: The default method in PySCF with DIIS enabled, as similar to most of the quantum chemistry software.
- **Full Online SCF**: Algorithm 1 without the adaptive mode switching mechanism. Online SCF is applied throughout the whole iteration process, with a learning rate of $10^{-2}$. To avoid the explosion of update interval $I_\Sigma^t$ when approaching to convergence, we simply set an upper limit of 10,000 for $I_\Sigma^t$.
- **Adaptive Online SCF**: Algorithm 1 including the adaptive mode switching mechanism. Regular SCF is allow to kick in when the iteration process is close to convergence. $T_{\text{cut-off}}$ and $T_{\text{cut-off-inc}}$ are set to be 100 and 10 respectively. $T_{\text{tabu}}$ is set to be 10.

The result is shown in Table 1. Compared with regular SCF approach, full online SCF method significantly reduced the number of nonconverged molecules in both Hartree-Fock and DFT scenarios, demonstrating its high convergence capacity in solving Equation (8). However, the gradient-like update rule of online methods results in comparatively low precision. This not only increases the number of required iterations significantly, but also restricts it from achieving higher convergence capacity under strict convergence criteria. The adaptive mode switching mechanism successfully resolved the issue. By allowing regular SCF to kick in at a later stage, with the flexibility to return back to online mode when oscillation occurs, adaptive online SCF achieves converged solution for all test molecules, with a moderate increase of average iteration number.

The behavior of adaptive online SCF is shown in Figure 6. While most of the molecules get converged with only one mode transition in Figure 6a, there are also a few "hard cases" like Figure 6b that require multiple mode transitions between online and regular SCF. The detailed statistics is shown in Table 2. The capability of mode switching is essential for the convergence capacity, since it provides unlimited chances of trials to reach a converged trajectory, which is lacking in regular SCF methods with only a few choices of starting point to select.

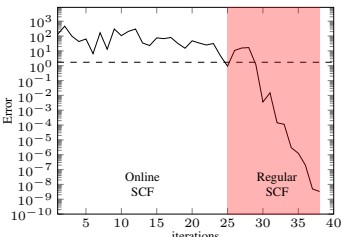
(a) Converge with one mode transition.

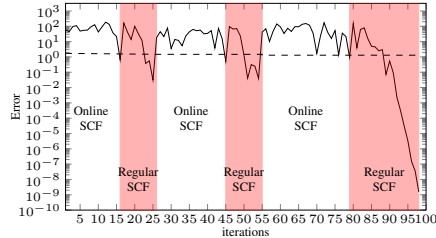
(b) Converge with multiple mode transitions.

Figure 6: Examples of converge curves for adaptive online SCF.

| #(mode transition) | 1 | 2 | 3 | 4 | 5 | 6 | 7 | 8 | > 8 |
|---|---|---|---|---|---|---|---|---|---|
| Hartree-Fock | 1143 | 179 | 13 | 2 | 0 | 0 | 1 | 0 | 0 |
| DFT with B3LYP | 1025 | 189 | 60 | 27 | 18 | 6 | 6 | 3 | 4 |

Table 2: Mode transition statistics of adaptive online SCF on sampled QM9 dataset. While all molecules finally converged on both Hartree-Fock and DFT scenarios and most of the them only require one mode transition from online to regular, the distribution of mode transition for DFT is more long-tailed (the 4 molecules with #(mode transition) > 8 have 12, 13, 15 and 32 transitions respectively).

## 5 Conclusion

In this work, we take the first steps in devising PCA-based algorithms for converging non-linear equations, and demonstrate successful applications in SCFs. This work contributes to both the field of computational science and machine learning as follows:

- For computational science, this work presents a new algorithm to converge SCF equations in electronic structure calculation with high successful rate, especially without any heuristics based on prior quantum mechanism knowledge to bootstrap the solving stage.
- For machine learning, this work explores a brand new area of "self-consistent" eigenvalue problems, especially SCF equations, for online PCA methods such as Oja's algorithm and EigenGame, which are previously regarded as specialized methods for $k$-PCA. While such methods can properly handle data stochasticity, this work shows that they are also capable of handling self-consistency, which leads to a potential of application in a broader field of scientific computing.

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
