# 1 Analytical Derivation of the Convergence Behavior in the Case Study

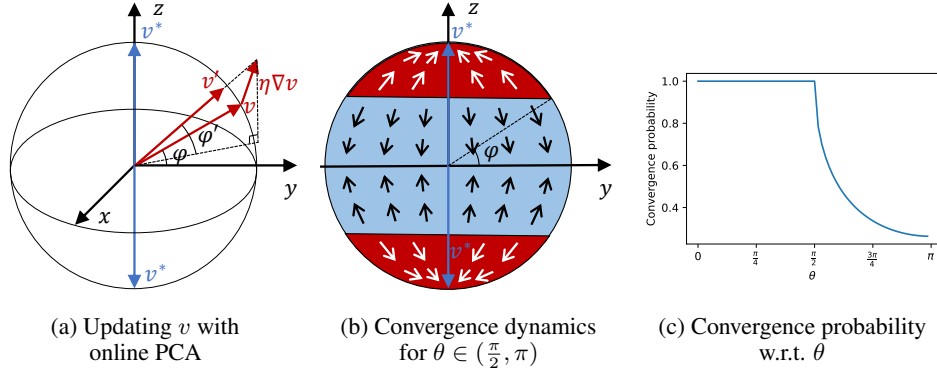

(a) Updating $v$ with online PCA

(b) Convergence dynamics for $\theta \in (\frac{\pi}{2}, \pi)$

(c) Convergence probability w.r.t. $\theta$

Figure 1: Convergence analysis of our proposed online PCA method for solving Equation (5) in the main paper. Note that vanilla fixed-point iteration can never converge in this case.

In this section, we provide the analytical derivation of the convergence behavior in Section 3.4 of the main paper.

For the online PCA method, here we select the simplest method, Oja's algorithm [3, 4]. Given a random sample $x_t \sim \mathcal{P}(x; \Sigma_t)$ at time step $t$, its update rule is as follows:

$$v_t = v_{t-1} + \eta(x_t x_t^\top v_{t-1}) \text{ and } v_t \leftarrow \frac{v_t}{\|v_t\|} \tag{1}$$

in which $\Sigma_t = F(v_t)$ is the covariance matrix at time step $t$.

For the convergence analysis, we explore the moving direction of $v$ during the iterative process (1). For $v = [0, \cos\varphi, \sin\varphi]^\top$, we have

$$\nabla v = F(v)v = Avv^\top Av = \begin{bmatrix} -\sin\theta\cos^3\varphi - \sin\theta\cos\varphi\sin^2\theta \\ \cos^2\theta\cos^3\varphi + \cos\theta\cos\varphi\sin^2\varphi \\ \cos\theta\cos^2\varphi\sin\varphi + \sin^3\varphi \end{bmatrix}$$

Then, we investigate whether the updated vector $v' = \frac{v+\nabla v}{\|v+\nabla v\|}$ will be closer to the ground-truth solution $v^*$, i.e., the angle between $v'$ and the $xy$-plane ($\varphi'$ in Figure 1a) will be larger than $\varphi$. That is

$$\frac{v_z'}{\sqrt{(v_x')^2 + (v_y')^2}} > \frac{v_z}{v_y}$$

which is equivalent to the following quadratic inequality w.r.t. $\cos^2\varphi$ for $\varphi \in (0, \frac{\pi}{2})$:

$$a\cos^4\varphi + b\cos^2\varphi + c > 0$$
$$a = 1 - \cos^2\theta + 2\cos^3\theta - \cos^4\theta$$
$$b = -4 + 2\cos\theta + \cos^2\theta - 2\cos^3\theta$$
$$c = 3 - \cos^2\theta$$

which implies

$$\varphi \in \begin{cases} (0, \frac{\pi}{2}], & \theta \in (0, \frac{\pi}{2}) \\ (\arccos\sqrt{x_1}, \frac{\pi}{2}], & \theta \in [\frac{\pi}{2}, \pi) \end{cases} \tag{2}$$

where $x_1 = (-b - \sqrt{b^2 - 4ac})/2a$.

## 2 Physical Background of Equation (8) in the Main Paper

In this section, we briefly introduce the physical background of Equation (8) in Section 3.5 of the main paper. We refer to [6, 2] for more details.

In computational physics, a foundation problem is to solve the Schrödinger equation of a many-body quantum system

$$H \left| \Psi \right\rangle = E \left| \Psi \right\rangle$$

where $H$ is the Hamilton operator for a system of $M$ nuclei and $N$ electrons described by their coordinates $R_A$ and $r_i$. With the Born-Oppenheimer approximation (nuclei are much heavier than electrons), we consider the electrons to be moving in the field of fixed nuclei, thus the kinetic energy of the nuclei (approximated as zero) and the repulsion between the nuclei (approximated to be constant) can be neglected. In this case we write $H$ as

$$H = \underbrace{-\sum_{i=1}^{N} \frac{1}{2} \nabla_i^2}_{\substack{\text{kinetic energy of the} \\ \text{electrons}}} \underbrace{-\sum_{i=1}^{N} \sum_{A=1}^{M} \frac{Z_A}{|r_i - R_A|}}_{\substack{\text{coulomb attraction be-} \\ \text{tween electrons and nu-} \\ \text{clei}}} + \underbrace{\sum_{i=1}^{N} \sum_{j=i+1}^{N} \frac{1}{|r_i - r_j|}}_{\substack{\text{repulsion between elec-} \\ \text{trons}}}$$

and $\Psi(r_1, \cdots, r_N)$ the electronic wave function, which should be normalized (i.e., $\langle \Psi | \Psi \rangle = \int \Psi^*(r_1, \cdots, r_N) \Psi(r_1, \cdots, r_N) dr_1 \cdots dr_N = 1$). For convenience, let operator $h(i) = -\frac{1}{2} \nabla_i^2 - \sum_{A=1}^{M} \frac{Z_A}{|r_i - R_A|}$ so $H$ can be rewritten as $H = \sum_{i=1}^{N} h(i) + \sum_{i=1}^{N} \sum_{j=i+1}^{N} \frac{1}{|r_i - r_j|}$. According to the variational principle, the problem of finding solution $\Psi$ for the ground state energy $E_0$ can be transformed to the following constrained optimization problem

$$\min \left\langle \Psi \right| H \left| \Psi \right\rangle \quad s.t. \left\langle \Psi | \Psi \right\rangle = 1$$

Since the multi-electron wave function $\Psi(r_1, \cdots, r_N)$ is computationally intractable when $N$ is large, a basic way is to approximate it as the product of $N$ orbital wave function $\psi_1(r_1)\psi_2(r_2) \cdots \psi_N(r_N)$ satisfying $\langle \psi_i | \psi_i \rangle = \int \psi_i^*(r) \psi_i(r) dr = 1, \forall i \in 1 \cdots N$, which is called "Hartree approximation". We also expand $\psi_i(r) = \sum_{u=1}^{K} C_{ui} \phi(r)$ as a linear combination of $K$ "basis functions" which is fixed and given, in this case our task becomes to determine the value of all $C_{ui}$ so as to determine the approximated wave function. To construct a Lagrange multiplier

$$L = \left\langle \Psi \right| H \left| \Psi \right\rangle - \sum_{i=1}^{N} \epsilon_i (\langle \psi_i | \psi_i \rangle - 1)$$

we have

$$\left\langle \Psi \right| H \left| \Psi \right\rangle = \sum_{i=1}^{N} \int \Psi^*(r_1, \cdots, r_N) h(i) \Psi(r_1, \cdots, r_N) dr_1 \cdots dr_N$$

$$+ \int \Psi^*(r_1, \cdots, r_N) \sum_{i=1}^{N} \sum_{j=i+1}^{N} \frac{1}{|r_i - r_j|} \Psi(r_1, \cdots, r_N) dr_1 \cdots dr_N$$

$$= \sum_{i=1}^{N} \int \psi_i^*(r) h(i) \psi_i(r) dr + \frac{1}{2} \sum_{i=1}^{N} \sum_{j \neq i}^{N} \int \psi_i^*(r_i) \psi_j^*(r_j) \frac{1}{|r_i - r_j|} \psi_i(r_i) \psi_j(r_j) dr_i dr_j$$

$$= \sum_{i=1}^{N} \sum_{u,v} C_{ui} C_{vi} \int \phi_u^*(r) h(i) \phi_v(r) dr$$

$$+ \frac{1}{2} \sum_{i=1}^{N} \sum_{j \neq i}^{N} \sum_{u,v,\lambda,\sigma} C_{ui} C_{\lambda j} C_{\sigma j} C_{vi} \int \phi_u^*(r_i) \phi_\lambda^*(r_j) \frac{1}{|r_i - r_j|} \phi_\sigma(r_j) \phi_v(r_i) dr_i dr_j$$

and

$$\sum_{i=1}^{N} \epsilon_i (\langle \psi_i | \psi_i \rangle - 1) = \sum_{i=1}^{N} \epsilon_i \left( \int \psi_i^*(r) \psi_i(r) dr - 1 \right)$$

$$= \sum_{i=1}^{N} \epsilon_i \left( \sum_{u,v} C_{ui} C_{vi} \int \phi_u^*(r) \phi_v(r) dr - 1 \right)$$

Let $H_{uv} = \int \phi_u^*(r)h(i)\phi_v(r)dr$, $S_{uv} = \int \phi_u^*(r)\phi_v(r)dr$ and $E_{uv\lambda\sigma} = \int \phi_u^*(r_1)\phi_\lambda^*(r_2)\frac{1}{|r_1-r_2|}\phi_\sigma(r_2)\phi_v(r_1)dr_1dr_2$ which are all constants since $\{\phi_i\}$ are given functions, we have

$$L = \sum_{i=1}^{N}\sum_{u,v}C_{ui}C_{vi}H_{uv} + \frac{1}{2}\sum_{i=1}^{N}\sum_{j\neq i}^{N}\sum_{u,v,\lambda,\sigma}C_{ui}C_{\lambda j}C_{\sigma j}C_{vi}E_{uv\lambda\sigma}$$

$$- \sum_{i=1}^{N}\epsilon_i(\sum_{u,v}C_{ui}C_{vi}S_{uv} - 1)$$

$$\frac{\partial L}{\partial C_{ui}} = 2\sum_{v}C_{vi}(H_{uv} + \sum_{j(\neq i)}\sum_{\lambda,\sigma}C_{\lambda j}C_{\sigma j}E_{uv\lambda\sigma} - \epsilon_i S_{uv})$$

Let $\frac{\partial L}{\partial C_{ui}} = 0, \forall i = 1\cdots N, u = 1\cdots K$ and we have

$$\sum_{v}C_{vi}(H_{uv} + \sum_{j(\neq i)}\sum_{\lambda,\sigma}C_{\lambda j}C_{\sigma j}E_{uv\lambda\sigma}) = \epsilon_i\sum_{v}C_{vi}S_{uv}, \forall i = 1\cdots N, u = 1\cdots K$$

Let $P_{\lambda\sigma} = \sum_{j(\neq i)}C_{\lambda j}C_{\sigma j}$, $[U_{\text{eff}}(P)]_{uv} = \sum_{\lambda,\sigma}P_{\lambda\sigma}E_{uv\lambda\sigma}$, $F_{uv} = H_{uv} + [U_{\text{eff}}(P)]_{uv}$ and $\Lambda = \text{diag}(\epsilon_1,\cdots,\epsilon_N)$, $P, U_{\text{eff}}(P), F \in \mathbb{R}^{K\times K}$, we have the matrix form of the above equation

$$FC = SC\Lambda$$

which is a generalized eigenvalue problem where the matrix $F$ to be decomposed is defined by the eigenvectors $C$.

Actually, there are several improved approximation theories based on the Hartree approximation mentioned above, in which the definitions of $P$ and $U_{\text{eff}}(P)$ are different. A most influential one is the Hartree-Fock theory in which the multi-electron wave function is approximated by a slater determinant

$$\Psi(x_1,\cdots,x_N) = \frac{1}{\sqrt{N!}}\begin{vmatrix} \chi_1(x_1) & \chi_2(x_1) & \cdots & \chi_N(x_1) \\ \chi_1(x_2) & \chi_2(x_2) & \cdots & \chi_N(x_2) \\ \vdots & \vdots & \ddots & \vdots \\ \chi_1(x_N) & \chi_2(x_N) & \cdots & \chi_N(x_N) \end{vmatrix}$$

to conform to the antisymmetry principle $\Psi(\cdots, x_i, \cdots, x_j, \cdots) = -\Psi(\cdots, x_j, \cdots, x_i, \cdots)$ whose famous representation is Pauli exclusion principle. In this way $P_{uv} = 2\sum_{i}^{N/2}C_{\lambda j}C_{\sigma j}$ (or in matrix form, $P = 2CC^\top$) and $[U_{\text{eff}}(P)]_{uv} = \sum_{\lambda,\sigma}P_{\lambda\sigma}E_{uv\lambda\sigma} - \frac{1}{2}\sum_{\lambda,\sigma}P_{\lambda\sigma}E_{u\lambda\sigma v}$, in which an "exchange term" $-\frac{1}{2}\sum_{\lambda,\sigma}P_{\lambda\sigma}E_{u\lambda\sigma v}$ is added.

## 3 Self-Consistent Field (SCF) method

Self-Consistent Field (SCF) is a standard method to solve Equation (8) in the main paper. An initial density matrix $P_0$ is generated via heuristics based on prior quantum mechanism knowledge, then the generalized eigen-decomposition problem $F(P_{t-1})V_t = SV_t\Lambda_t$ is repeatedly solved to obtain eigenvectors $V_t$ and density matrix $P_t = 2V_tV_t^\top$ at the $t$-th iterations, $t = 1, 2, \cdots$, until $|P_t - P_{t-1}|$ is less than the convergence threshold. The detailed routine is shown in Algorithm 1.

To solve the generalized eigen-decomposition problem, it should be transformed to standard form first. To achieve this, the orthogonalization technique introduced in [6] is applied to eliminate the overlap matrix $S$. First, find a linear transformation $X$ so that $X^\top SX = I$. There are several ways to achieve this, a popular one is named "canonical orthogonalization" which lets $X = U\text{diag}(s^{-1/2})$, where $U$ and $s$ are all eigenvectors and eigenvalues of $S$. Note that all eigenvalues of $S$ are positive so there is no difficulty of taking square roots. Then let $V' = X^{-1}V$ ($V = XV'$), we have $F(V)XV' = SXV'\Lambda$. Multiply $X^\top$ on the left and we have $X^\top F(V)XV' = X^\top SXV'\Lambda$. Let $F'(V) = X^\top F(V)X$, we have

$$F'(V)V' = V'\Lambda \tag{3}$$

which is a standard eigen-decomposition problem.

---

**Algorithm 1** Self-Consistent Field (SCF) method

---

**Input:** $H, S, U_{\text{eff}}(\cdot)$ in Equation (8) and (9) in the main paper. Converge threshold $\epsilon$.
**Output:** $V^*$, the solution of Equation (8)
  Obtain initial density matrix $P_0$ via initial guess methods (e.g., SAD or core Hamiltonian).
  Find $X$ satisfying $X^\top S X = I$.
  $t \leftarrow 0$
  **while** $|P_t - P_{t-1}| < \epsilon$ **do**
    $F_t \leftarrow H + U_{\text{eff}}(P_t)$
    $F'_t \leftarrow X^\top F_t X$
    $t \leftarrow t + 1$
    Obtain precise eigenvectors $V'_t$ of $F'_{t-1}$ corresponding to the top-$k$ smallest eigenvalues via classical eigen-decomposition methods such as QR iteration.
    $V_t \leftarrow X V'_t$
    $P_t \leftarrow 2 V_t V_t^\top$
  **end while**
  $V^* \leftarrow V_t$

---

# 4 Other Numerical Adaptations to Solve Equation (8) in the Main Paper

In this section, we provide some additional numerical adaptations applied in Section 3.5 of the main paper to converge Equation (8).

**DIIS**: Not only the principal components, but also the covariance matrix can be updated incrementally. For solving SCF equations, especially in the context of electronic structure calculation, the direct inversion of the iterative subspace (DIIS) method is empirically effectively and widely applied as default. The basic idea is to update $\Sigma_t$ at iteration $t$ as a linear combination of matrices in previous iterations $\Sigma_{t-T_{\text{subspace}}}, \cdots, \Sigma_{t-1}$ as follows in which $T_{\text{subspace}}$ is the number of subspaces

$$\Sigma_t = \sum_{i=t-T_{\text{subspace}}}^{t-1} c_i \Sigma_i \quad \left( \sum_{i=t-T_{\text{subspace}}}^{t-1} c_i = 1 \right) \tag{4}$$

in which $c_i$ is the mixing coefficients, which are required to add to one. The value of $c_i$ is determined by a constrained optimization minimizing a specifically constructed error, whose detail can be found in [5]. In the experiments involving electronic structure calculation, we apply DIIS on all tested methods.

**Momentum**: The selection of learning rate $\eta$ for online PCA is highly tricky for different $N$ and $F(\cdot)$ in Equation (8). To enhance the robustness of the algorithm towards learning rate, we introduce the momentum method for the update of $V$ as follows:

$$M_t = \beta M_{t-1} + \eta \nabla V_{t-1}, \tag{5}$$
$$V_t = V_{t-1} + M_t, \tag{6}$$

in which $\nabla V_{t-1}$ is the update direction computed by online PCA, and $\beta$ is the momentum term.

**Sample-free Update**: Note that for some online PCA algorithms here could be slightly modified to be integrated into the above model more efficiently, exploiting the fact that the change of distribution is explicitly given by the reconstruction function $F(\cdot)$.

Given a distribution $\mathcal{P}_t$ at a certain time step $t$, to update the top principal component $v_t$ with online PCA, the standard approach is to sample one or a mini-batch of data from $\mathcal{P}_t$, then perform online PCA on the sampled data to update $v_t$ in an incremental way based on $v_{t-1}$. However, if the covariance matrix of $\mathcal{P}_t$ is already given as $\Sigma_t$ explicitly, the sampling process may not be required for many of the online PCA algorithms, since they also infer the covariance matrix from the sampled data. For example, in Oja's algorithm [3, 4], given a sampled data $x_t \sim \mathcal{P}_t$ at time step $t$, the top principal component $v_t$ is updated as:

$$v_t \leftarrow v_{t-1} + \eta(x_t x_t^\top v_{t-1}) \text{ and } v_t \leftarrow \frac{v_t}{\|v_t\|}$$

Note that $x_t x_t^\top$ is actually an estimation of covariance matrix $\Sigma_t$ of $\mathcal{P}_t$. If $\Sigma_t$ is explicitly given, the above process can be simplified as

$$v_t \leftarrow v_{t-1} + \eta(\Sigma_t v_{t-1}) \text{ and } v_t \leftarrow \frac{v_t}{\|v_t\|}$$

## 5    Results on Other Dataset

We also did experiments on other datasets, however, preliminary results shows that small datasets like W4-17[1] are not challenging enough for our experiment, as both our methods and the baseline converges very well.