# OpenReview forum: "Online PCA in Converging Self-consistent Field Equations"
_NeurIPS.cc/2023/Conference — NeurIPS 2023 poster_

### Official Review · Reviewer_uFr2 · 2023-07-03

**Soundness:** 3 good
**Presentation:** 3 good
**Contribution:** 2 fair
**Rating:** 6
**Confidence:** 4

**Summary:**

This paper explores online PCA methods for a certain type of non-linear eigenvalue problem.

**Strengths:**

This is a well-written paper on an interesting problem in computational science.

**Weaknesses:**

My main critique is that, as presented, the contribution of this paper appears not as much to machine learning or data science, rather to computational science.




**Questions:**

Could you please expand the last paragraph of the conclusion section on how this is a contribution to machine learning?

**Limitations:**

-

---

> ### Author Rebuttal · Authors · 2023-08-10
>
> Thank you for the review. Since our work lies in the applications of online PCA methods to the otherwise unknown area and the specific self-consistent Eigen problem, our work contributes to the machine learning community by expanding the reach of online PCA methods. Before our work, online PCA methods are regarded as specialized methods to handle the stochasticity issue in online or streaming environment. Our work shows that they are also capable of handling self-consistency issues in solving an important class of nonlinear eigenvalue problem. Additionally, the concept of self-consistency is closely related to mean-field theory, which is an active topic in machine learning. Machine learning and physics are closely connected with mutual concepts such as mean-field and Boltzmann machine, which leads to a potential application of our work on a boarder range of machine learning applications involving mean-field characteristics.
>
> Moreover, according to the CfP of NeurIPS 2023,
>
> > NeurIPS 2023 is an interdisciplinary conference that brings together researchers in machine learning, neuroscience, statistics, optimization, computer vision, natural language processing, life sciences, natural sciences, social sciences, and other adjacent fields.
>
> and we believe that our work fits in the topics of "Optimization" and "Machine learning for sciences".

---

> > ### Comment · Reviewer_uFr2 · 2023-08-12
> > **response to authors**
> >
> > Since the rest of the reviewers rather liked the paper, and I have no concern regarding its quality, I have raised my score.
> >
> > I do encourage the authors to mention clearly in the paper (possibly in the introduction and not at the end of the paper) the merits of the paper pertaining to machine learning and data science.

---

### Official Review · Reviewer_myiv · 2023-07-04

**Soundness:** 3 good
**Presentation:** 3 good
**Contribution:** 3 good
**Rating:** 7
**Confidence:** 3

**Summary:**

The paper presents a new online-PCA based algorithm with some additional computational innovations to solve self-consistent systems. They add a mode-switching method and delayed calculation to improve convergence issues. The results are very good, but on a somewhat limited/niche dataset.

**Strengths:**

The paper is well presented, simple and shows very good results on a task that is seemingly impossible to solve with other methods.

**Weaknesses:**

I am not an expert in the application area (electronic structures), but it seems like there could have been more extensive experiments to illustrate the benefits of the method. I don't think sec 4.1 gives a good enough picture of the benefits of the new method.

Also, the algorithm boxes on page 4 and 5 are slightly confusing and figure 4 is misplaced and is covering up some text it seems (at least in my printed version).

**Questions:**

Is the methods that is being compared with in sec 4.2 the only other methods that can solve the SCF problems?

Are there any other real-world problems other than electronic structures where this methodology can be used? Non-stationary time series?

In Figure 2. Is it possible to explain the intuition behind the role of F? It would help the paper to have an idea what the "typical role" of F in systems such as these is. (maybe this is not possible, but I still ask...)

What is DIIS? I did not see the introduction to this and it seems like it suddenly popped up without a definition.

**Limitations:**

The limitations are adequately addressed.

---

> ### Author Rebuttal · Authors · 2023-08-10
>
> Thank you for your constructive comment. The detailed responses regarding each concern are listed below.
>
> For other methods, while multiple existing methods exists, most of them are some variations of the DIIS technique. An incomplete list includes energy-DIIS, augmented-DIIS, LIST, GDIIS and RMM-DIIS, which may behave more efficiently or with better convergency in specific scenarios, and can be good alternatives when standard DIIS fails. However, most of these variations are about including quantum-chemistry-specific information into the DIIS procedure (e.g., energy-DIIS minimizes the Hartree-Fock energy functional, and augmented-DIIS minimizes augmented Roothaan Hall energy function), and these ideas can also be included in our methods. To compare apples to apples, we should also develop corresponding variations of our method from “energy-Adaptive SCF”, “augmented-Adaptive SCF” to “RMS-Adaptive SCF” to make a fair comparison, which could be a bit too exhausted and quantum-chemistry oriented, and may not be of interest to a majority of NeurIPS audiences. We leave works about how these variations affect our proposed method and DIIS as a future work to be presented to quantum chemistry communities.
>
> We will correct the misplacement of Figure 4 in the revised paper, sorry for the confusion. The covered sentence is "where $\psi_0$ is the initial angle between the vector and the xy plane."
>
> For other problems, we think it may also model the crowd behavior in social science, in which the behavior of an individual will be influenced by the "representative options" of the crowd, and the representative options of the crowd will be influenced by averaging all the change of the individuals. Generally speaking, systems with "mean-field" characteristics may be benefitted from this methodology.
>
> For the role of F, while F is the "input" of the problem in eq (1) which could be arbitrarily defined, a typical definition of F in the scenario of electronic structure is defined in eq (9) whose computational detail is shown in the line 41 and 51 in the appendix.
>
> For DIIS, it is an convergence acceleration technique introduced in line 71 of the appendix. Sorry for the confusion and we will revise the main paper to include an introduction of DIIS.

---

### Official Review · Reviewer_EC9k · 2023-07-10

**Soundness:** 2 fair
**Presentation:** 3 good
**Contribution:** 2 fair
**Rating:** 6
**Confidence:** 3

**Summary:**

In this work, the authors approach solving the Self-consistent Field (SCF) equation from a principal component analysis (PCA) for non-stationary time series perspective. They shows that, the equilibrium state of such an online PCA corresponds to the solution of the SCF equations. By doing so, this work is abled to achieve better convergence compared to the traditional fixed-point iteration methods for solving such equations.

**Strengths:**

- As mentioned in the paper, solving self-consistent Field (SCF) equation is of great significance in computational science for its connection to the Schrödinger equation. So proposing a novel approach, to overcome the non-convergence issues of the traditional fixed-point iteration methods for solving such equations, is important.
- The authors also mentioned that, this is the first steps in devising PCA-based algorithms for converging non-linear equations. So further study in this direction can help solving other such relevant problems.

**Weaknesses:**

- Please edit line 181 in the manuscript. Some part of the line is omitted by figure 4.

**Questions:**

- Why only the first Eigen vector?

**Limitations:**

- This paper focuses on solving one important but rather niche problem.

---

> ### Author Rebuttal · Authors · 2023-08-09
>
> Thank you for your constructive comment.
>
> We will correct the layout error in line 181. The covered sentence is "where $\psi_0$ is the initial angle between the vector and the xy plane."
>
> The reason to include only the first eigenvector/eigenvalue in eq (1) is for the simplicity of form, as the main issue of SCF equations -- "self-consistency" is remained. In a more complicated case shown in eq (8), we include top-k eigenvector/eigenvalues and discussed this extension in line 211 and 212.

---

### Official Review · Reviewer_h8kr · 2023-07-23

**Soundness:** 2 fair
**Presentation:** 3 good
**Contribution:** 3 good
**Rating:** 5
**Confidence:** 3

**Summary:**

This paper proposes a new method for solving self-consistent field equations - a form of nonlinear generalized eigenvalue problem in which the matrix being diagonalized is a function of the eigenvectors of the diagonalization. These equations are of great interest in quantum chemistry, and are typically solved via fixed-point iteration, but can struggle with the stability of the iterative process.
This paper proposes a connection to the Principal Component method (PCA) by viewing the function F(v) as a mapping from a vector to a data distribution, wherein the map F acts as a form of decoder/reconstruction function and PCA itself acts as an encoder/compression function. This formulation is entirely equivalent to the original problem, but allows the use of certain modified online/adaptive PCA methods to stabilize the iterative procedure.

The authors demonstrate that this method performs superior to vanilla SCF iterations in a specific, theoretically-tractable case study, then apply the method to the more difficult case of solving the Kohn-Sham equations in electronic structure theory. The authors test their method on the QM9 dataset, which contains a large number of molecules for the purposes of electronic structure calculations. They sample 1% of the dataset at random, then evaluate their method on each case in question and compare to the existing SCF interpretation from the PySCF package. They find that their method results in convergence for all molecules considered (compared to 70-90%), while requiring roughly 2-3x more iterations.

**Strengths:**

The proposed method is interesting, and applies machine learning techniques to a foundational problem in quantum chemistry. Improving the accuracy or efficiency of Hartree-Fock/DFT calculations would be highly valuable to the quantum chemistry community, and is thus an interesting area of research.
The proposed method performs well on the theoretical case study, and the results in table 1 show a clear improvement in convergence rate over the existing baselines under discussion.

**Weaknesses:**

It would be helpful to have a better understanding of the significance of these results in the context of quantum chemistry, where they are most likely to be used. The results shown in Table 1 show that the existing baseline achieves a convergence rate of approximately 70-90% depending on the scenario, whereas the proposed approach achieves 100% convergence at a cost of approximately 2-3x the number of iterations. Within the field of quantum chemistry, is failure to converge a significant limitation, and is this tradeoff worth it? While I am not a quantum chemist, my understanding is that DFT calculations are already exceptionally computationally expensive, and significantly increasing the number of iterations required for convergence may be a very severe drawback.

In addition, it would be good to see a more thorough comparison. The experiments in this paper are only performed on a single dataset, and only compare to a single baseline. While I, again, am not a quantum chemist, my brief review of the literature revealed a number of existing methods seeing widespread use - including RMM-DIIS (is this the one used as a baseline in the paper?) as well as Davidson or Blocked Davidson iterations, and combination methods combining both RMM-DIIS and Blocked Davidson. Unless I am misunderstanding the applicability of these methods to the problem under consideration, it would be helpful to see a comparison to a broader range of baselines, as well as a wider range of datasets. As is, I feel like the comparisons in section 4 are too narrow to provide a compelling case for the proposed approach, but it is entirely possible that I am missing important context from the field of quantum chemistry. I am happy to revisit this if my understanding is incomplete.

**Questions:**

Is convergence rate a primary limiting factor in Hartree-Fock/DFT calculations in practice? If so, how does this compare with the potential downsides of increased computational cost required by your method?

Am I correct in assuming that the RMM-DIIS method is the one used as a baseline in the paper? If so, was there a reason other methods were omitted from comparisons? Is there a reason these other methods are not applicable to the problem under consideration?

**Limitations:**

It would be helpful if the authors gave a more complete presentation of the applicability of the proposed approach in the context of the quantum chemistry literature, and the significance of their results in that context.

---

> ### Author Rebuttal · Authors · 2023-08-09
>
> Thank you for your constructive comment. The detailed responses regarding each concern are listed below.
>
> For the trade-off between convergence and efficiency, note that such a trade-off characteristic can be adjusted in our proposed method by setting different $T_{\text{cut-off}}$ in L230 of the paper. If a larger $T_{\text{cut-off}}$ is set, then the portion of Online SCF will be more dominant, which leads to empirically better convergence with the cost of larger number of iterations. If a smaller $T_{\text{cut-off}}$ is set, then the portion of Regular SCF will be more dominant, which leads to smaller number of iterations for converged molecules, while non-converged molecules are more possible to appear (if $T_{\text{cut-off}}$ is set to be zero, the the proposed method is equal to Regular SCF). In this work, since we are more focused on the convergence side as the title suggests, we set a relatively large $T_{\text{cut-off}}$ to boost the convergence performance. However, the parameter can be set differently to fit practical scenarios. If you are aware that the molecule of your interest may be challenging to get a converged solution, you may wish to set a larger $T_{\text{cut-off}}$ to strengthen the convergence capability of the method, while otherwise you can set a smaller $T_{\text{cut-off}}$ for efficiency. We will elaborate more about this feature of our method in the revised paper.
>
> For the comparison, while multiple existing methods exists, most of them are some variations of DIIS technique. An incomplete list includes energy-DIIS, augmented-DIIS, LIST, GDIIS and RMM-DIIS you mentioned, which may behave more efficiently or with better convergency in specific scenarios, and can be good alternatives when standard DIIS fails. However, most of these variations are about including quantum-chemistry-specific information into the DIIS procedure (e.g., energy-DIIS minimizes the Hartree-Fock energy functional, and augmented-DIIS minimizes augmented Roothaan Hall energy function), and these ideas can also be included in our methods. To compare apples to apples, we should also develop corresponding variations of our method from “energy-Adaptive SCF”, “augmented-Adaptive SCF” to “RMS-Adaptive SCF”, which could be a bit too exhausted and quantum-chemistry oriented, and may not be of interest to a majority of NeurIPS audiences. We leave works about how these variations affect our proposed method and DIIS as a future work to be presented to quantum chemistry communities.
>
> For the dataset in Sec 4.2, we select QM9 dataset mainly for its sufficient challenge to convergence. Since the focus of our work is on the convergence capability, we intend to set the experiment to be challenging enough to differentiate different methods. We actually did experiments on other datasets, however, preliminary results shows that small datasets like W4-17 [1] are not challenging enough for our experiment, as both our methods and the baseline converges very well. We will mention the result on other datasets in the revised paper.
>
> [1] Karton, Amir, Nitai Sylvetsky, and Jan M. L. Martin. “W4-17: A Diverse and High-Confidence Dataset of Atomization Energies for Benchmarking High-Level Electronic Structure Methods.” Journal of Computational Chemistry 38, no. 24 (2017): 2063–75.

---

> ### Comment · Reviewer_h8kr · 2023-08-21
> **Response to Authors**
>
> After seeing the response from the authors, many of my concerns remain unaddressed.  Given that the rest of the reviewers had a higher opinion of the work I will raise my rating to borderline, but I still believe that the experimental comparisons are incomplete, and that the benefit of the proposed approach within the context of quantum chemistry compared to existing approaches is unclear. I do think the paper is interesting and has potential, but as is I don't think the paper makes a clear enough case for an improvement relative to the state of the art.

---

### Comment · Area_Chair_B7sr · 2023-08-11
**Discussion period**

Dear reviewers and authors,

Thank you very much for your work on this submission and its evaluation. Now that the authors have responded to the reviews, I *strongly encourage* the reviewers to acknowledge the review, to look at other reviews and rebuttals for this submission, and to adjust their scores if needed. Thanks to those that have already done so.

Authors have the possibility to reply if further questions are needed, until the 16th.

Thank you very much to all,

Area Chair

---

### Decision · Program_Chairs · 2023-09-21

**Decision:**

Accept (poster)

**Comment:**

There is an overall consensus among reviewers that this is an interesting submission that should be accepted, which is also my evaluation.